# Study of the Spectral Characteristics of Crops of Winter Wheat Varieties Infected with Pathogens of Leaf Diseases

**DOI:** 10.3390/plants13141892

**Published:** 2024-07-09

**Authors:** Roman Danilov, Oksana Kremneva, Igor Sereda, Ksenia Gasiyan, Mikhail Zimin, Dmitry Istomin, Alexey Pachkin

**Affiliations:** 1Federal State Budgetary Scientific Institution «Federal Research Center of Biological Plant Protection» (FSBSI FRCBPP), Krasnodar 350039, Russia; gasiyankkk@mail.ru (K.G.); istomin.dmitriy.94@mail.ru (D.I.); capricorn-53@yandex.ru (A.P.); 2Department of Cartography and Geoinformatics, Faculty of Geography, M. V. Lomonosov Moscow State University, Moscow 119991, Russia; iisereda@mail.ru (I.S.); ziminmv@mail.ru (M.Z.)

**Keywords:** ground-based spectrometry, winter wheat, wheat diseases, spectral characteristics

## Abstract

Studying the influence of the host plant genotype on the spectral reflectance of crops infected by a pathogen is one of the key directions in the development of precision methods for monitoring the phytosanitary state of wheat agrocenoses. The purpose of this research was to study the influence of varietal factors and disease development on the spectral characteristics of winter wheat varieties of different susceptibility to diseases during the growing seasons of 2021, 2022 and 2023. The studied winter wheat crops were represented by three varieties differing in susceptibility to phytopathogens: Grom, Svarog and Bezostaya 100. Over three years of research, a clear and pronounced influence of the varietal factor on the spectral characteristics of winter wheat crops was observed, which in most cases manifested itself as an immunological reaction of specific varieties to the influence of pathogen development. The nature of the influence of the pathogenic background and the spectral characteristics of winter wheat crops were determined by the complex interaction of the development of individual diseases under the conditions of a particular year of research. A uniform and clear division of the spectral characteristics of winter wheat according to the intensity of the disease was recorded only at a level of pathogen development of more than 5%. Moreover, this gradation was most clearly manifested in the spectral channels of the near-infrared range and at a wavelength of 720 nm.

## 1. Introduction

Winter wheat is one of the leading agricultural crops grown worldwide. Economically important diseases of wheat include pathogens of septoria spp. and brown spot (*Pyrenophora tritici-repentis* (Died.) Drechsler), as well as pathogens that cause powdery mildew (*Blumeria graminis* (DC.) Speer), brown rust (*Puccinia triticina* Erikss.) and yellow rust (*Puccinia striiformis* West.). These diseases are widespread in the world [1,2]. In Russia, there are also cases of wheat affected by these diseases. This is especially pronounced in the southern regions [3,4,5]. According to FAO and the UN, wheat diseases cause annual losses. Thus, in developed countries, crop losses of up to 10% are observed and, in developing countries, up to 20–50% [6]. In Russia, losses of grain crops of up to 25–35% are caused by rust fungi, septoria, yellow spot, powdery mildew, fusarium and root rot [7,8].

One of the factors for successful plant protection is the ability to quickly monitor large areas of agricultural land. This approach can provide high quality data [9,10]. Vast cultivated areas make it difficult to conduct phytosanitary monitoring using traditional visual accounting methods. Consequently, there is a lack of proper control by specialists and, therefore, there is an urgent need for a fundamental scientific and methodological ground for early diagnosis of the main pathogens of wheat diseases based on aerospace information technology, as well as high-performance ground-based measurements. The current level of Earth remote-sensing equipment is characterized by the emergence of space- and aviation-based hyper-spectral equipment. Researchers believe that determining the spectral properties of an object in specific narrow wavelength ranges creates the opportunity to identify hidden special changes [11,12,13,14,15,16,17]. It is not possible to record such changes on the ground without special tools and special methods.

Many researchers study the physiological state of wheat crops through hyper-spectral data analysis [18,19,20,21,22,23,24,25,26,27,28,29,30,31,32,33,34]. The most significant results have been obtained by identifying diseases such as fusarium head blight (*F. graminearum* Schwabe) [19,20,21,22]. Further research is needed to determine the spectral properties of different wheat varieties since these properties may distort the overall picture of biotic and abiotic stress in the wheat canopy. Thus, Delwiche et al., 2000, studying the possibilities of detecting fusarium head blight in three different wheat varieties, established a significant influence of the variety factor on the adequacy of pathogen-detection models. Models developed for only one variety were found to be useless when applied to other varieties [19]. Zhang et al. studied the relationship between nutritional stress and yellow rust disease in three wheat varieties. They identified the only disease-sensitive plant growth index PhRI [24,25]. Others have established the possibility of using hyper-spectral data for phenotyping the resistance of different wheat varieties affected by septoria [32,33]. The results of studying the influence on the spectral properties of different varieties and variety mixtures of pathogens causing winter wheat yellow spot have also been obtained [34].

Thus, studying the influence of the host plant genotype on the spectral reflectance of crops, as well as on the biochemical and physiological characteristics of plants exposed to pathogen infection, is one of the key directions in the development of precision methods for monitoring the phytosanitary state of wheat agrocenoses. Currently, there is a growing trend in the number of scientific papers on this topic. However, a complete picture and unambiguous results on the spectral parameters of plants have not been obtained. This is attributed to the poor reproducibility of the results due to the lack of experimental data in the field of reflective properties of different varieties of the same species throughout years or seasons of research.

Here we aim to study the influence of variety factors and disease intensity on the spectral characteristics of winter wheat varieties of different susceptibility during the growing seasons of 2021, 2022 and 2023.

Accordingly, the following tasks were set:(1)Study the nature of similarities and differences in the spectral characteristics of crops of winter wheat varieties in individual years of research;(2)Study the relationship between the spectral characteristics of winter wheat crops and disease intensity in individual years of research;(3)Study the relationship between spectral characteristics and disease intensity of each individual variety during three years of research;(4)Assess the possibility of differentiating crops according to the degree of pathogen development.

## 2. Results

Using two-factor analysis of variance, how the spectral characteristics of winter wheat crops are affected by varietal differences, as well as the intensity of various diseases (Table 1), was established. However, the nature of their mutual influence was not entirely clear during the three years of research. Thus, in 2021, only a separate influence of crop varietal characteristics and disease development indicators was observed. In 2022 and 2023, observations were made of the influence of plant varietal characteristics and the development of disease on the plant.

According to 2022 data, the full influence of variety factors and disease development, as well as their combination, was manifested in the spectral channels of 575, 700, 1445, 2035 and 2295 nm. In the spectrum channels of 550, 660, 720 and 1725 nm, the influence of individual varieties and their combination with the disease development factor was observed. At wavelengths of 920, 1085, 1135, 1245, 1285 and 2345 nm, which belong to the near- and mid-infrared spectral ranges, only the influence of the variety factor was observed.

In 2023, the influence of varietal factors and the development of diseases on winter wheat crops had the greatest visibility. This effect was especially strong at wavelengths 490, 520, 550, 575, 660, 700, 720, 1675 and 1725 nm. A separate influence of the two factors, similar to in 2021, was observed at mid-infrared wavelengths of 845, 920, 1085, 1135, 1215, 1245, 1285 and 1445 nm.

A comparative post-hoc (Duncan) analysis showed (Figure 1, Table 2) that, during three years of research, despite the difference in the level of development and pathogenic composition, a clear and pronounced separation of control and infected crops was noticed. Moreover, in 2021, this division manifested itself in the form of an excess of the average values of the crops SBC in control plots over the infected ones in a number of spectral channels.

On the contrary, infected plots of the studied varieties of winter wheat exceeded the control ones in terms of reflectivity in 2022 and 2023. At the same time, the most informative spectral channels in which the difference between control and infected plots appeared are wavelengths of 575, 660, 700 and 2035 nm. A distinctive feature of the 2021 studies was the lowest development of pathogens compared to those in 2022 and 2023, as well as the simultaneous manifestation of yellow and brown rust on crops in infected plots. One could assume that this is what determined the nature of the differences between the control plots and the infected ones.

It is important to note that over the entire three-year study, the influence of varietal characteristics had an advantage in influencing the spectral properties of winter wheat crops. The development of the pathogenic background affected varieties only in accordance with the influence of varietal characteristics. In addition, it was found that the nature of the similarities and differences in the crops of compared winter wheat varieties according to spectral characteristics in the same growth phase, but in different years of research, was also ambiguous.

Differences only appeared in the mid-infrared spectral channel in 2021 and 2023. These years of research were characterized by leaf rust in the pathogenic background. In the spectral channels 490, 520, 550, 845, 1675, 1725 and 2345 nm, differences appeared only in 2023, when the maximum level of pathogen development was observed. In 2022, differences were noted in the 1445 nm spectral channel. This, in all likelihood, can be associated with the characteristics of composition and intensity of the pathogenic background for a given period of time.

The average level of development of the entire pathogenic background in control plots in 2021 was 0.33% and, in the infected ones, 0.72%. The difference between them was 0.54%. In 2022, this was 1.07 and 1.52%, respectively, with a difference of 0.45%. In 2023, the average level of development of all pathogens in the control background was 1.7%, and in the infected background 6.8%. The difference between them was 5.03%.

Thus, we can state that the factor of separability of winter wheat crops according to spectral characteristics with a minimum difference in the development of pathogens was 0.5%. In individual spectral channels of 490, 520, 845 and 2345 nm, variations appeared with a difference in the pathogenic background of 5%.

In 2021, the difference in the spectral characteristics of infected and control areas of the studied winter wheat crops was determined by the mutual influence of the development of yellow and brown rusts. According to the results of the correlation analysis for yellow and brown rusts, a negative relationship was revealed between the indicators of their development and the variable values of the spectral brightness coefficient of the spectral channels of the visible and near-infrared ranges (Table 3). Also, for both pathogens, a high and statistically significant correlation was established in the mid-infrared spectral channel of 1445 nm. Separately, for yellow rust, a high and statistically significant correlation was identified in the 2035 nm spectral channel.

Tan spot and septoria were allocated to another group. A high and statistically significant correlation was also identified between these pathogens. The development indicators of powdery mildew were characterized by an average level of correlation with the development indicators of tan spot and septoria. No significant level of correlation was found with variable SBC values of spectral channels for tan spot and septoria.

A comparative post-hoc analysis showed that the difference in the spectral characteristics of the studied winter wheat crops with different levels of pathogen development was largely determined by the influence of the variety factor. Thus, in 2021, the main and most pronounced difference between control crops and infected spots was identified. It was determined by the presence of the joint manifestation of yellow and brown rusts on the latter. Despite the fact that the distribution and development of brown spot and septoria blight were high, the influence of these pathogens on the spectral response of the studied winter wheat crops was not revealed.

The pathogenic background of 2022 was characterized by the highest level of yellow rust manifestation over the entire research period, which negatively correlated with the development of septoria (Table 5). Correlation analysis revealed that yellow rust development indicators were characterized by a high and positively directed relationship with variable values of the spectral channels of the visible range.

Correlation analysis revealed that yellow rust was characterized by a positive correlation of the development degree with variable SBC values in the visible spectral channels, and the highest and statistically significant correlation in the 720 nm spectral channel. Septoria stood out with an average (0.5–0.6) level of correlation in the spectral channels of the mid-infrared range (1445, 1675, 1725, 2035, 2295 and 2345 nm), and powdery mildew in the near-infrared spectral range (Table 3).

A comparative post-hoc analysis indicated that the most pronounced and significant difference was characterized by infected crops of the Bezostaya 100 and Svarog varieties. This was manifested by the highest SBC values for almost all spectrum channels (Appendix A, Table A1). It is important that the spectral properties of two different varieties, when considered separately, are very different, despite the same level of pathogenic background. In general, the most pronounced differentiation of crops according to the degree of development of yellow rust appeared in the 720 nm spectral channel. Differentiation of crops according to the degree of septoria development in most of the spectral channels was subject to the influence of the variety factor. Categories of crops with pathogen development of 0.5–1% were identified only at wavelengths of 1445 and 2345 nm.

In 2023, the highest level of disease development was observed compared to 2021 and 2022, respectively, which was manifested in the strongest impact on the spectral properties of the studied winter wheat crops (Table 5). Against a pathogenic background, a statistically significant negative correlation was observed between the development of powdery mildew and leaf rust. In addition to these pathogens, a positive correlation of development was observed in septoria with powdery mildew, as well as a negative correlation in septoria with leaf rust.

The mutual influence of the three pathogens manifested in the correlation between their development and the variable SBC values of the near-infrared spectral channels (Table 3). This relationship was highest and statistically significant for powdery mildew. A statistically significant relationship between the developments of leaf rust and variable SBC values appeared at a wavelength of 1445 nm.

A comparative post-hoc analysis demonstrated that the best differentiation of crops according to disease intensity was also revealed in the spectral channels of the near-infrared range (Appendix A, Table A2). At the same time, a tendency for SBC values to increase along with the growing intensity of disease development was observed for powdery mildew and septoria. On the other hand, for leaf rust, a decrease in SBC values with an increase in the severity of disease development was observed.

Thus, according to the intensity of powdery mildew, the studied crops were clearly divided into groups with corresponding indicators of disease intensity of 0–3, 4, 10–12 and 30%, but according to the degree of development of septoria by 0, 2–3 and 4%. Solely crops with an indicator of 0% stood out according to the intensity of leaf rust. When grouping SBC values obtained at a wavelength of 1445 nm, no statistically significant differentiation of crops according to the degree of leaf rust intensity was revealed, despite the high level of correlation with the development of the disease in this wavelength range.

There was no significant correlation found in any of the tan spot spectral channels, although in 2023 it reached an average level of development of 18.37%. The division of crops according to disease intensity in agreement with the Duncan criterion was also extremely ambiguous. Apparently, this was due to the very specific reaction of each specific variety to the influence of a given pathogen.

The most pronounced differentiation of crops according to generalized groups appeared in the 2345 nm spectral channel. At the same time, crops of the first category with minimal pathogenic background, as well as of the fifth and sixth categories of crops with maximum indicators of disease development, were identified. The differentiation of the second and third categories was determined by the variety factor, which was superimposed by the influence of pathogens of various levels of development and composition. However, upon closer examination, it is possible to distinguish between these categories based on combinations of differences in SBC values in various spectral channels.

As a result, tables were created that compare spectral properties between individual varieties over a three-year study period (Table 4, Appendix A, Table A3, Table A4 and Table A5).

Against the background of pathogenic development on crops of the Bezostaya 100 variety, a statistically significant positive correlation of the development of powdery mildew with yellow spot was revealed, which influenced the appearance for this variety of a statistically significant correlation between the external signs of powdery mildew damage and the SBC values of the visible and near-infrared wavelength ranges (Table 4).

**Table 4 plants-13-01892-t004:** Results of the assessment of the correlation between the degree of disease development and variable values of the spectral brightness coefficient of winter wheat crops of the varieties Bezostaya 100, Svarog and Grom for the research period 2021–2023.

Pathogen	Spectral Range, nm												
	490	550	660	700	720	845	1245	1445	1675	2005	2035	2295	2345
Bezostaya 100													
Powdery mildew	0.84 *	0.84 *	0.84 *	0.84 *	0.84 *	0.61	0.81 *	0.41	0.75	0.46	0.58	0.32	0.46
Yellow spot	0.78	0.78	0.78	0.78	0.78	0.34	0.44	0.14	0.37	−0.03	0.14	−0.1	−0.03
Septoria	0.31	0.31	0.31	0.31	0.31	0.03	0.14	−0.14	0.03	−0.26	−0.26	−0.49	−0.43
Yellow rust	−0.46	−0.46	−0.46	−0.46	−0.46	0.15	0.21	0.03	0.21	0.7	0.39	0.39	0.58
Brown rust	−0.65	−0.65	−0.65	−0.65	−0.65	−0.65	−0.65	−0.65	−0.65	−0.39	−0.65	−0.65	−0.65
Generalized categories	0.66	0.66	0.66	0.66	0.66	0.71	0.89 *	0.37	0.77	0.6	0.6	0.31	0.54
Svarog													
Powdery mildew	0.52	0.52	0.52	0.81 *	0.52	0.26	0.52	0.52	−0.81 *	0.81 *	0.12	0.06	−0.31
Yellow spot	0.7	0.7	0.7	0.52	0.7	0.94 *	0.7	0.7	−0.39	0.52	0.82	0.52	-
Septoria	0.66	0.66	0.66	0.43	0.66	0.77	0.66	0.66	−0.49	0.43	0.43	0.03	−0.66
Yellow rust	−0.88 *	−0.88 *	−0.88 *	−0.52	−0.88 *	−0.7	−0.88 *	−0.88 *	0.03	−0.52	−0.64	−0.52	0.54
Brown rust	−0.13	−0.13	−0.13	−0.39	−0.13	0.39	−0.13	−0.13	0.39	−0.39	0.39	0.13	-
Generalized categories	0.49	0.49	0.49	0.77	0.49	0.31	0.49	0.49	−0.83 *	0.77	0.14	0.03	0.2
Grom													
Powdery mildew	−0.58	−0.72	−0.67	−0.52	−0.72	−0.23	−0.64	−0.41	−0.67	0.17	−0.12	−0.72	−0.81 *
Yellow spot	0.38	0.75	0.55	0.81 *	0.75	−0.46	0.06	0.93 *	0.55	−0.64	0.12	0.75	0.41
Septoria	−0.09	−0.09	−0.03	−0.31	−0.09	0.26	0.26	−0.43	−0.03	0.09	−0.09	−0.09	0.37
Yellow rust	−0.52	−0.76	−0.52	−0.88 *	−0.76	0.21	−0.21	−0.88 *	−0.52	0.27	−0.27	−0.76	−0.21
Brown rust	0.1	0.51	0.34	0.68	0.51	−0.78	−0.3	0.85 *	0.34	−0.54	0.34	0.51	0.17
Generalized categories	−0.37	−0.14	−0.09	−0.09	−0.14	−0.66	−0.49	0.03	−0.09	−0.14	0.37	−0.14	0.09

Notes: *—statistical significance of data correlation is confirmed.

A comparative post-hoc analysis showed that crops of the Bezostaya 100 variety were well differentiated by the degree of development of powdery mildew, tan spot and septoria into groups with indicators of 0–1 and 2–10% in the spectral channels of the visible and near-infrared ranges (Appendix A, Table A3). At the same time, there was a tendency for SBC values to increase with the intensification of disease manifestation.

The pathogenic background of crops of the Svarog variety during three years of research was characterized by a positive interaction between the development of powdery mildew, tan spot and septoria, which negatively correlated with yellow rust manifestation. A high level of statistically significant and negatively directed correlation of variable SBC values of most spectral channels with symptoms of yellow rust development was revealed for crops of the Svarog variety (Table 4). A positive, statistically significant correlation of variable SBC values was revealed for powdery mildew in the 700 and 2035 nm spectral channels, for tan spot in the 845, 920 and 1085 nm spectral channels, and for septoria in the 920 nm spectral channel.

A comparative post-hoc analysis showed that the clearest and most pronounced division of crops was also revealed when grouping variable SBC values corresponding to the degree of intensity of yellow rust (Appendix A, Table A4). Crops of the Svarog variety were clearly divided according to the degree of development of yellow rust with a gradation of 0, 1–2 and 6%. At the same time, crops with a lower level of pathogen development were characterized by the highest average values. That is, the opposite trend was observed for powdery mildew, tan spot and septoria on crops of the Bezostaya 100 variety. Only groups with maximum pathogen development rates were identified when dividing crops by degree of intensity of powdery mildew, tan spot and septoria.

For the pathogenic background of the Grom variety, a positive and statistically confirmed relationship between the development of tan spot and brown rust was revealed, which in turn was negatively correlated with yellow rust. For indicators of brown spot and yellow rust on crops of the Grom variety, a characteristic was determined—a significant dependence at wavelengths of 700 and 1445 nm (Table 4). A statistically significant high correlation for leaf rust appeared only at a wavelength of 1445 nm, and for powdery mildew in the 2345 nm spectral channel.

A comparative post-hoc analysis showed that the differentiation of crops of the Grom variety was ambiguous according to the development degree of pathogens and the intensity of powdery mildew, tan spot and septoria (Appendix A, Table A5). Groups of crops with maximum pathogen development rates of 2–3% were identified for brown and yellow rust.

A cumulative comparison of the spectral characteristics of all three varieties over the entire period of research revealed a statistically significant relationship between only two pathogens: tan spot and yellow rust. A statistically significant correlation with the variable values of the visible and mid-infrared spectral channels was established for these two pathogens (Appendix A, Table A6).

## 3. Discussion

### 3.1. Assessment of the Mutual Influence of Variety Factors and Disease Development on the Spectral Characteristics of Winter Wheat Crops

Using two-factor analysis of variance, the influence of varietal factors and disease development on the spectral characteristics of the studied winter wheat crops was established (Table 1). In 2021, only a separate influence of crop varietal characteristics and disease development indicators was observed. In 2022 and 2023, an interaction between varietal factors and disease development was observed. This interaction of factors manifested itself in the form of differences in the spectral characteristics of the infected and control crops of each individual variety in the spectral channels of the visible and mid-infrared spectral ranges. In the near-infrared range, there was no interaction be-tween variety factors and disease development. Selected visible spectral channels (490, 520, 550, 575, 660 and 700 nm) are sensitive to pigment content (chlorophyll, carotenoids), phytomass, canopy structure, moisture content and plant stress. The mid-infrared range contains areas of water absorption, as well as areas sensitive to plant stress, lignin and starch content [12]. Thus, it can be assumed that the interaction of variety factors and disease development was a stress response of the host plant. This was accompanied by changes in the pigment composition of plants, disruption of the plant canopy structure, as well as disruption of water and temperature conditions [14,15].

A comparative post-hoc (Duncan) analysis showed (Table 2) that over three years of research, despite the difference in intensity and composition of pathogens, a clear and pronounced separation of control and infected crops was observed.

It was revealed that in the 575, 660, 700 and 2035 nm spectral channels, these differences appeared annually, regardless of the conditions and the specific year of research. At the same time, the manifestation of variability at other wavelengths of the spectral range depended on the composition and disease intensity in the pathogenic background. It is known that the 575 nm green spectral channel is sensitive to the content of plant pigments [11]. The 660 nm wavelength of the red part of the spectrum is associated with chlorophyll absorption and depends on many factors (phytomass, crop, canopy structure, nitrogen content, moisture content and stress) [12]. The 700 nm range is also associated with chlorophyll absorption and is most sensitive to changes in overall plant health. The light wavelength of 2035 nm corresponds to the maximum absorption of water in the mid-infrared region of the spectrum [13,14].

### 3.2. Comparison of the Spectral Characteristics of Crops of the Studied Varieties across the Time Period of Each Individual Year of Research

When considering the spectral properties of the studied winter wheat crops across the time period of each individual year, the following patterns can be identified:-The nature of the influence of the pathogenic background on the spectral characteristics of winter wheat crops was determined by the complex interaction of disease development in a specific year of research. For instance, in 2021, the differences in the spectral characteristics of the control and infected backgrounds were determined by mutual manifestations of yellow and brown rusts in the latter. In 2022, the greatest impact was exerted by a negative correlation between the development of yellow rust and septoria, and in 2023 powdery mildew and brown rust;-A clear and pronounced influence of varietal characteristics on the spectral properties of winter wheat was observed over three years. In most cases, this manifested itself as an immunological reaction of a particular variety to the influence of pathogen development. Different cultivars with similar pathogen indicators often exhibited strong differences in spectral response;-A regular and clear division of the spectral properties of winter wheat crops according to the intensity of the disease was observed only at a level of pathogen development of more than 5%. Moreover, this gradation was clearest in the spectral channels of the near-infrared range and at a wavelength of 720 nm. The most pronounced differentiation of crops according to generalized groups appeared in the 2345 nm spectral channel.

However, upon closer examination, it was possible to identify distinctions in these categories based on combinations of differences in SBC values in various spectral channels.

-When the pathogenic background was below 5%, as in 2022 and 2023, the reflectivity of crops was largely determined by the influence of the variety factor or the interaction of the variety factor and disease intensity;-Tan spot showed no significant correlations with SBC variables, even at its highest level in 2023. In all likelihood, this pathogen is the most variety-specific, i.e., it is determined most by the immunological reaction of a particular variety;-A high level of correlation with variable SBC values of the 1445 nm spectral channel was revealed for leaf rust in 2021 and 2023.

### 3.3. Comparison of the Spectral Characteristics of Each Individual Variety over Three Years of Research

A comparison of the spectral properties of each individual variety over three years of research revealed the following patterns:-A fairly clear correlation and differentiation of spectral characteristics according to the development degree of individual pathogens was revealed for crops of each individual variety;-The spectral properties of each individual variety were determined by the different direction of the correlation relationship of disease intensity indicators in the general pathogenic background. Moreover, the nature of this relationship was different for the compared varieties, even with similar indicators of external disease manifestation. For instance, a pronounced and statistically significant correlation of external signs of powdery mildew manifestation with variable values of SBC spectral channels of the visible and near-infrared ranges appeared for the Bezostaya 100 variety. Such a relationship was not found in crops of the Svarog variety, though they were characterized by the highest rates of powdery mildew intensity. But a high level of statistically significant and negatively directed correlation of variable SBC values of most spectral channels with symptoms of yellow rust development was revealed. Moreover, the gradation of yellow rust development indicators in the Bezostaya 100 and Svarog varieties was almost identical;-The clearest and most consistent differentiation of the spectral characteristics of winter wheat varieties was manifested by pathogen intensity. Thus, crops of the Bezostaya 100 variety were well differentiated by the intensity of powdery mildew, tan spot and septoria into groups with indicators of 0–1 and 2–10%. Crops of the Svarog variety were clearly divided according to the intensity of yellow rust with a gradation of 0, 1–2 and 6%. Crops with minimum and maximum indicators were differentiated according to general categories of disease intensity.

### 3.4. Prospects for Further Development of Research

The international scientific literature presents works aimed at identifying the development of individual pathogens in different wheat varieties [18,19,20,21,22,23,24,25,26,27,28,29,30,31,32,33,34]. These works proved the possibility of diagnosing diseases using hyperspectral analysis methods. However, results obtained even within the study of a single disease vary widely. This difference in results is potentially explained by the biochemical characteristics of different wheat varieties, climate, as well as a complex combination of the influence of abiotic and biotic stress factors [13,14,15]. Research on varietal and biochemical differences forms only a very small part of all studies devoted to spectral studies of vegetation. In addition, there is no reference methodology or database on which the authors of the works could rely in their research.

A feature of these studies is a detailed study of the nature of the interaction of varietal factors and complex development of the main economically significant diseases on the pathogenic background of crops of three varieties of winter wheat in 2021, 2022 and 2023.

The research results allow us to conclude that it is necessary to accumulate and systematize data over a significant period of time in relation to specific wheat varieties. A solution to this problem may be to create a model based on long-term data. Such a model should contain parameters of the mutual influence of pathogens on a specific variety, taking into account the limiting weather factors of a particular year. It is also possible to create a generalized model that allows extrapolation of data for many varieties based on studying a group of reference varieties [32,33]. In addition, the importance of studying biochemical changes in plant tissues under the influence of pathogens should be recognized [35,36].

## 4. Materials and Methods

### 4.1. Organization of Test Plots and Experimental Design

We conducted the research in the experimental fields of the Federal Research Center of Biological Plant Protection (FRCBPP), Krasnodar (45° 2.413′ 0″ N, 38° 58.5598′ 0″ E, 29 m above sea level) in 2021–2023 (Figure 2). The Köppen climate classification scheme assigns the climate of the study area as transitional from temperate continental to subtropical (Cfa) [37]. This region is characterized by long, hot summers and mild to moderately warm winters. Transitional seasons are poorly expressed. The average annual precipitation is 700–750 mm. The average annual air temperature is +13.4 °C and the average annual air humidity is 71%. The soil cover of the territory is represented by leached chernozems with low humus [38].

The studied winter wheat crops were represented by three varieties bred by the National Grain Center named after. P. P. Lukyanenko (Krasnodar, Russia), which are susceptible to phytopathogens: Bezostaya 100, Svarog and Grom. Each plot was divided into two zones: 1—disease-protected by fungicides (clean background), 2—with an infectious background of pathogens. Artificial inoculation methods were used to develop brown and yellow rusts in the experimental area [39]. Inoculation of plants was carried out in the first ten days of April (HS phase 30–32). A mixture of urediniospores and talc in a ratio of 1:100 at a load of 5 mg spores/m^2^ was used as an inoculating agent. The development of pathogens causing yellow spot, septoria and powdery mildew occurred against a natural infectious background. The creation of a control background (without diseases) was carried out by 2-fold treatment with the systemic fungicide Sokol, KS: 1st treatment on 25–31 April (flag-leaf phase), 2 May. 10–15 (phase “beginning of flowering” GS 61).

The research methodology was based on a comparative analysis of high-precision ground-based spectrometric measurements with the results of field phytopathological studies (Figure 3).

### 4.2. Field Experiments

The main period for conducting research was the beginning of the intensive appearance of leaf-stem diseases. This period fell in the second ten days of May (phase GS 61 “beginning of flowering”). This time period was the leading link for creating a predictive model of pathogen development since it allows for a comparative analysis of quantitative indicators of pathogen development. This analysis took into account the development of the pathogen from primary symptoms (after the incubation period) to intensive manifestation, taking into account the influence of varietal factors and weather conditions of a particular year.

The degree of development of the disease was assessed using the method of visual calculation of the ratio of the proportion of the affected area of the plant leaf blade to its total area (Figure 4). Visual observations of the development of winter wheat diseases were carried out while moving along the diagonal of each experimental plot with an area of 10 m^2^. A total of 30 plants were selected for analysis. After this, for each tier (first, second leaf, etc.) a percentage assessment of leaf damage was given according to international scales. The degree of damage from rust diseases was assessed using the Peterson scale [40]; the degree of pyrenophorosis damage was assessed using the modified Saari–Prescott scale [41]; the degree of damage by powdery mildew and septoria was assessed using a special scale developed by CIMMYT [42].

Analyzing the test areas, the average degree of disease development was calculated using the following Formula (1):(1)$$R=\frac{1}{n}\sum_{i=1}^{n}r_{i}$$
where R is the average degree of disease development, %; r—degree of the disease development of an individual plant, %; n—total number of registered plants, pcs.

Areas with crops of the studied varieties were divided into general categories in accordance with the average indicators of pathogen development (Table 5).

### 4.3. Ground-Based Spectrometric Measurements

Ground-based spectrometry was carried out remotely at a height of 1.2–1.4 m from the earth’s surface in the range of electromagnetic radiation from 350 to 2500 nm, with a spectral resolution of 1–10 nm. For this purpose, the ASD FieldSpec 3 Hi-Res spectroradiometer (Boulder, CO, USA) was used [43], which is designed for field remote sensing of the environment. The device has a non-removable fiber optic cable with factory calibration, thanks to which a high signal-to-noise ratio is achieved, which in turn ensures high accuracy of results for better identification and analysis of materials. To ensure comparability of the obtained data, measurements were carried out on days with clear sunny weather with a minimum amount of clouds. The sun’s altitude was more than 35°.

Such analysis conditions were chosen due to the fact that, under such circumstances, lighting conditions change significantly less. This period of time and weather conditions reduce the possible error associated with the tilt of the sun. To analyze the spectral properties of the vegetation cover, two series of measurements of five repetitions were carried out. In the intervals between measurements, the panel reflecting light was calibrated. This decision was made due to the need to reduce the influence of uneven lighting. Vegetation cover was measured along the diagonal of the experimental plot, which corresponded to the methodology for conducting field surveys of plants for the presence of pathogens.

The results of ground-based spectrometric measurements are a set of spectral brightness coefficient (SBC) values. These values indicate the degree to which sunlight was reflected from plant surfaces at each wavelength.

**Table 5 plants-13-01892-t005:** Average indicators of disease development in winter wheat crops of varieties Bezostaya 100, Svarog and Grom at the beginning of intensive manifestation of all leaf diseases in the GS 60–70 “blooming” phase in the growing seasons 2021–2023.

Variety	Experience Option	Powdery Mildew	Septoria	Yellow Tan	Yellow Rust	Brown Rust	General Categories
2021							
Bezostaya 100	Control	0.01	0.8	0	0	0	1
Svarog		0.01	1	0	0	0	1
Grom		0.01	2.1	0.01	0	0	2
Bezostaya 100	Infected	0.01	1.9	0	0.3	0.3	4
Svarog		0.01	2.3	1.9	0.2	0.05	3
Grom		0	4.1	0.3	0.2	0	5
2022							
Bezostaya 100	Control	0.8	0.26	-	1.48	-	1
Svarog		4.02	1.87	-	2.21	-	3
Grom		2.82	2.06	-	1.06	-	2
Bezostaya 100	Infected	2	0.94	-	7.72	-	5
Svarog		4.42	0.87	-	6.41	-	5
Grom		1.76	4.2	-	0.78	-	4
2023							
Bezostaya 100	Control	3.43	1.23	1.87	-	0	2
Svarog		10.23	3.03	0.77	-	0	3
Grom		2.6	0	2.03	-	0.77	1
Bezostaya 100	Infected	12.93	3.77	10.03	-	0	4
Svarog		28.17	3.35	6.03	-	0	6
Grom		0	2.2	18.37	-	15.9	5

### 4.4. Data Processing

To identify specific spectral ranges indicating the manifestation of pathogenic changes, an analysis of changes in the morphology of reflective properties according to their actual state during field experiments was carried out.

Pre-processing of the analysis results and graphical visualization were carried out using the OriginPro 8.5.1 software package.

The pathogenic effect on the spectral properties of winter wheat plants in different wavelength ranges was assessed using two-way analysis of variance. To analyze the measurement results, the following wavelengths were selected: 490, 520, 550, 575, 660, 700, 720, 845, 1445, 1675 and 2345 nm. These spectral ranges are closely related to the biophysical characteristics of plants and are widely used in such studies [10]. Statistical processing of data from selected spectral channels was carried out with the calculation of the average value and standard deviation.

As a result of the analysis, the BCS values were grouped into categories corresponding to different degrees of the disease. The grouping of SBC values was carried out according to various parameters, including plant variety, damage from certain diseases, or compliance with three selected plant backgrounds. Correlation analysis of the relationship between the development of the disease and the number of detected pathogen spores was carried out on the basis of nonparametric statistical methods using Spearman’s correlation at a high significance level of 95%. All methods of statistical analysis were performed in the Statistica 2010 program.

## 5. Conclusions

A regular and clear distinction in the spectral characteristics of winter wheat according to disease intensity was observed only when the pathogen development level exceeded 5%. Moreover, this gradation was clearest in the spectral channels of the near-infrared range and at a wavelength of 720 nm. During three years of research, a strong influence of varietal characteristics on the spectral properties of winter wheat crops was discovered. In most cases, this manifested itself as an immunological reaction of a particular variety to pathogens.

The features of the pathogenic influence on the spectral properties of winter wheat crops was characterized by a complex interaction between the manifestations of individual diseases in a specific year of research. The reflectivity of crops was largely determined by the influence of the variety factor or the interaction of the variety factor and disease intensity when the pathogenic background was below 5%, as in 2022 and 2023.

## Figures and Tables

**Figure 1 plants-13-01892-f001:**
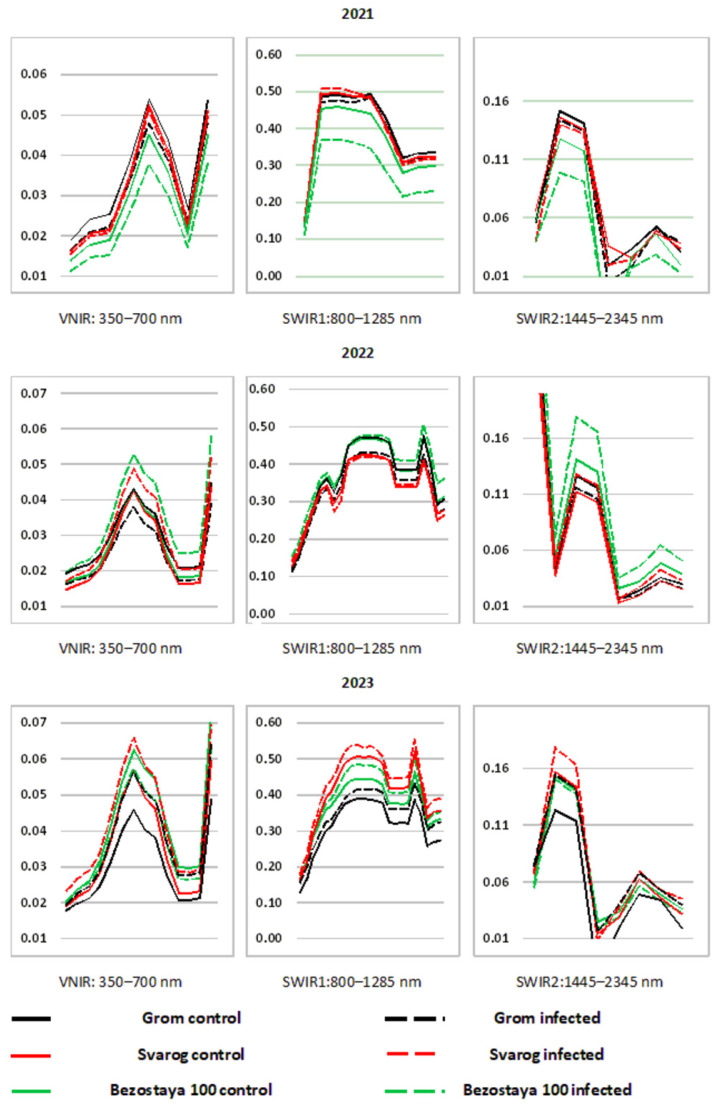
Spectral images of crops of studied varieties of winter wheat at the moment of the onset of intensive manifestation of all leaf diseases in the “flowering” phase of GS 60–70 in the growing season 2021–2023.

**Figure 2 plants-13-01892-f002:**
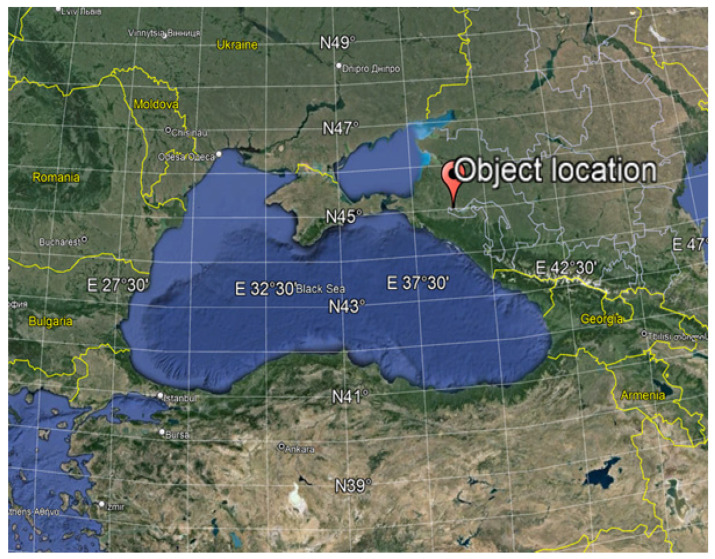
Geographical location of the research site.

**Figure 3 plants-13-01892-f003:**
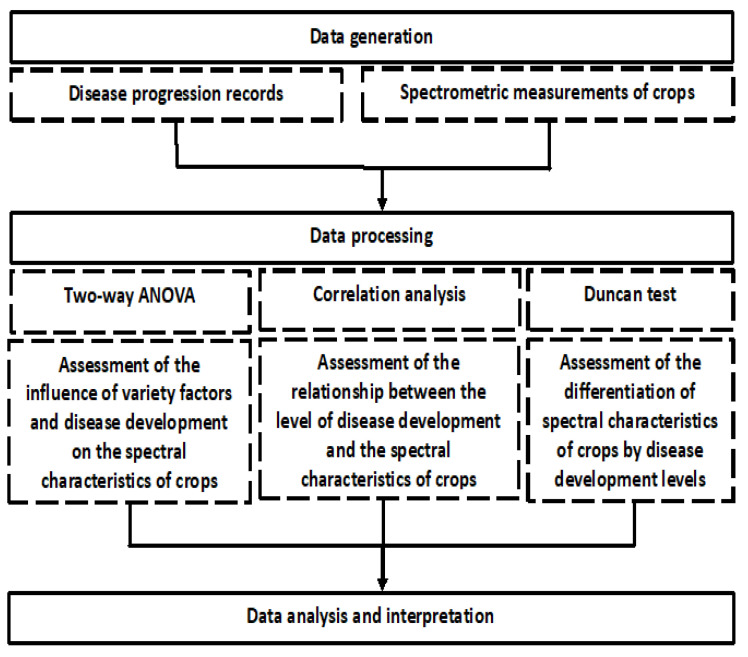
Research methodology diagram.

**Figure 4 plants-13-01892-f004:**
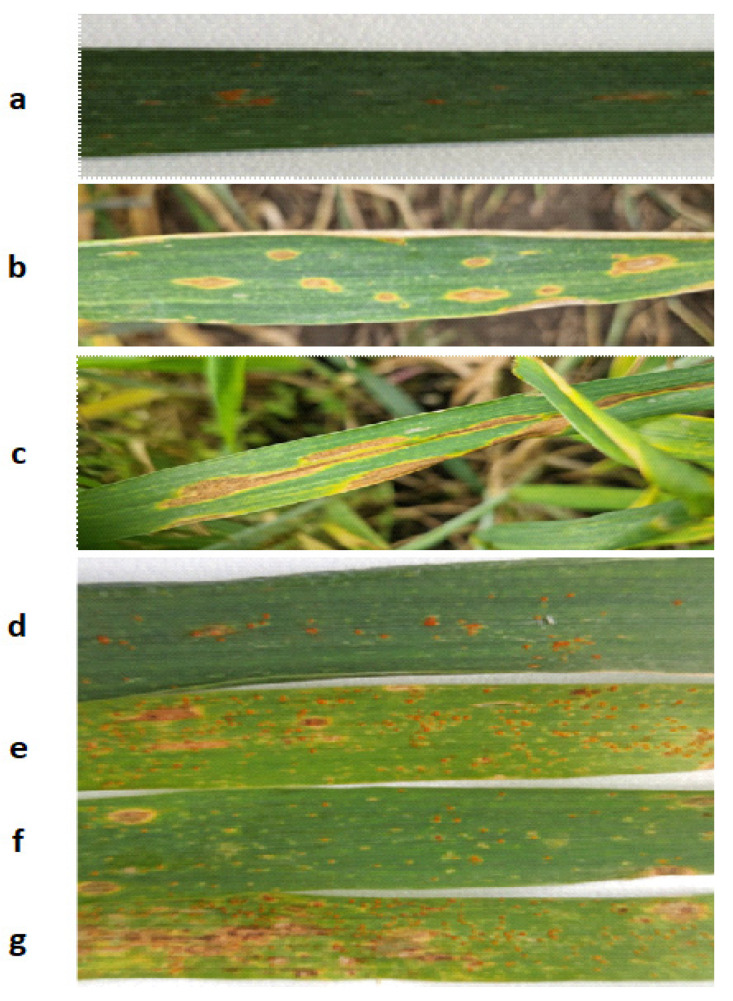
Symptoms and degree of development of diseases. (**a**) Brown rust—10%; (**b**) Tan spot—25%; (**c**) Septoria—40%; (**d**) Brown rust—15%; (**e**) Brown rust—25%, Tan spot—5%; (**f**) Brown rust—10%, Tan spot—5%; (**g**) Brown rust—30%, Tan spot—10%.

**Table 1 plants-13-01892-t001:** The influence of variety factors and disease development on the spectral brightness coefficient indicators of crops of the studied varieties of winter wheat at the time of the onset of intensive manifestation of all leaf diseases in the GS 60–70 “blooming” phase in the growing seasons 2021–2023.

Spectral Channels	Variety			Infectious Background			Variety * Infectious Background		
	2021	2022	2023	2021	2022	2023	2021	2022	2023
490	*	-	*	*	-	*	-	*	*
520	*	-	*	*	-	*	-	*	*
550	*	*	*	*	-	*	-	*	*
575	*	*	*	*	-	*	-	*	*
660	*	-	*	*	*	*	-	*	*
700	*	*	*	*	*	*	-	*	*
720	*	*	*	*	-	*	-	*	*
845	*	-	*	-	-	*	-	-	-
920	*	*	*	-	-	*	-	-	-
1085	*	*	*	-	-	*	-	-	-
1135	*	*	*	*	-	*	-	-	-
1215	*	*	*	*	-	*	-	-	-
1245	*	*	*	*	-	*	-	-	-
1285	*	*	*	*	-	*	-	-	-
1445	-	*	*	-	*	-	-	*	-
1675	*	*	*	*	-	*	-	*	*
1725	*	*	*	-	-	*	-	*	*
2005	*	*	-	-	-	-	-	-	-
2035	-	*	-	*	*	*	-	*	*
2295	*	*	-	-	*	-	-	*	-
2345	-	*	-	-	-	*	-	-	-

Notes: *—mathematically reliable influence of the factor on SBC indicators is confirmed.

**Table 2 plants-13-01892-t002:** A comparative post-hoc (Duncan) analysis of spectral brightness coefficient values of the studied winter wheat varieties in different spectral ranges at the time of the onset of intensive manifestation of all leaf diseases in the GS 60–70 “blooming” phase in the growing seasons 2021–2023.

Variety	Experience Option		Spectral Range, nm							
			490	550	660	720	845	1445	1675	2345
2021										
Bezostaya 100		Control	0.019 b	0.045 b	0.021 b	0.134 b	0.454 b	0.057 a	0.128 b	0.020 a
Svarog			0.021 bc	0.052 c	0.023 b	0.152 b	0.494 b	0.066 a	0.146 b	0.038 a
Grom			0.025 d	0.054 c	0.027 c	0.150 b	0.487 b	0.058 a	0.152 b	0.031 a
Bezostaya 100		Infected	0.015 a	0.038 a	0.017 a	0.111 a	0.367 a	0.039 a	0.098 a	0.012 a
Svarog			0.021 bc	0.051 bc	0.021 b	0.150 b	0.506 b	0.041 a	0.140 b	0.034 a
Grom			0.022 c	0.048 bc	0.023 b	0.136 b	0.472 b	0.056 a	0.144 b	0.039 a
2022										
Bezostaya 100		Control	0.018 ab	0.043 ab	0.018 a	0.128 ab	0.460 a	0.053 b	0.141 a	0.039 b
Svarog			0.016 a	0.042 ab	0.016 a	0.127 ab	0.419 a	0.037 a	0.112 a	0.025 a
Grom			0.021 b	0.043 ab	0.021 a	0.122 ab	0.463 a	0.043 ab	0.126 a	0.030 ab
Bezostaya 100		Infected	0.022 b	0.053 c	0.025 b	0.153 c	0.466 a	0.072 c	0.179 b	0.051 c
Svarog			0.019 ab	0.049 bc	0.021 a	0.140 bc	0.412 a	0.047 ab	0.127 a	0.033 ab
Grom			0.017 ab	0.038 a	0.017 a	0.113 ab	0.422 a	0.039 a	0.116 a	0.026 a
2023										
Bezostaya 100		Control	0.024 b	0.063 c	0.030 c	0.176 cd	0.440 b	0.059 a	0.153 b	0.036 a
Svarog			0.022 ab	0.057 b	0.023 a	0.164 bc	0.500 c	0.067 a	0.157 b	0.032 ab
Grom			0.020 a	0.046 a	0.021 a	0.128 a	0.383 a	0.074 a	0.123 a	0.020 ab
Bezostaya 100		Infected	0.024 b	0.057 b	0.027 b	0.164 bc	0.477 c	0.056 a	0.150 b	0.033 ab
Svarog			0.027 c	0.066 c	0.029 bc	0.180 d	0.536 d	0.069 a	0.178 c	0.045 c
Grom			0.023 b	0.056 b	0.028 bc	0.156 b	0.406 a	0.076 a	0.155 b	0.040 c

Notes: R—an indicator of the degree of progression of the disease; data represent the average mean value of the SBC and standard error in each column; the average values with the same letter do not differ significantly.

**Table 3 plants-13-01892-t003:** Results of assessing the correlation between the degree of disease development and variable values of the spectral brightness coefficient of spectral channels in the 2021–2023 research.

Pathogen	Spectral Range, nm												
	490	550	660	700	720	845	1245	1445	1675	2005	2035	2295	2345
2021													
Powdery mildew	−0.39	0.13	−0.39	0.13	0.13	0.13	−0.13	0.13	−0.13	0.13	0.39	−0.39	−0.65
Yellow spot	0.46	0.33	0.46	0.33	0.21	0.58	0.27	−0.27	0.27	0.33	−0.09	0.7	0.52
Septoria	0.49	0.2	0.49	0.2	0.09	0.31	0.26	−0.37	0.26	0.26	−0.37	0.66	0.54
Yellow rust	−0.46	−0.62	−0.46	−0.62	−0.62	−0.31	−0.62	−0.93 *	−0.62	−0.22	−0.93 *	−0.31	−0.15
Brown rust	−0.68	−0.51	−0.68	−0.51	−0.51	−0.17	−0.68	−0.85 *	−0.68	−0.1	−0.68	−0.51	−0.51
Generalized categories	0.03	−0.35	0.03	−0.35	−0.43	−0.23	−0.23	−0.72	−0.23	−0.17	−0.75	0.17	0.17
2022													
Powdery mildew	0.03	0.31	0.03	0.31	0.2	−0.54	−0.66	−0.31	−0.31	−0.66	−0.31	−0.14	−0.31
Yellow spot	-	-	-	-	-	-	-	-	-	-	-	-	-
Septoria	−0.2	−0.49	−0.2	−0.49	−0.77	0.09	−0.26	−0.66	−0.66	−0.26	−0.66	−0.77	−0.66
Yellow rust	0.43	0.77	0.43	0.77	0.94 *	0.03	0.14	0.54	0.54	0.14	0.54	0.71	0.54
Brown rust	-	-	-	-	-	-	-	-	-	-	-	-	-
Generalized categories	0.43	0.7	0.43	0.7	0.64	−0.12	−0.14	0.2	0.2	−0.14	0.2	0.32	0.2
2023													
Powdery mildew	0.49	0.71	0.14	0.14	0.71	0.94 *	0.94 *	−0.49	0.6	−0.14	−0.09	−0.14	0.2
Yellow spot	0.2	−0.09	0.14	0.14	−0.09	−0.2	−0.2	0.31	−0.09	0.09	0.6	0.26	0.49
Septoria	0.49	0.54	0.14	0.14	0.54	0.77	0.77	−0.49	0.43	0.2	0.26	−0.14	0.37
Yellow rust	-	-	-	-	-	-	-	-	-	-	-	-	-
Brown rust	−0.54	−0.78	−0.3	−0.3	−0.78	−0.78	−0.78	0.85 *	−0.3	−0.3	0.17	0.34	−0.07
Generalized categories	0.6	0.49	0.43	0.43	0.49	0.6	0.6	0.14	0.71	0.03	0.77	0.54	0.83 *

Notes: *—statistical significance of data correlation is confirmed.

## Data Availability

Data is contained within the article.

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
