# Peer review of "Study of the Spectral Characteristics of Crops of Winter Wheat Varieties Infected with Pathogens of Leaf Diseases"

_plants, 2024, doi:10.3390/plants13141892_

Round 1

Reviewer 1 Report

Comments and Suggestions for Authors

The study by Danilov et al (Plants-3072092), evaluated the use of ground-based spectrometry (range of electromagnetic radiation from 350 to 2500 nm) to verify the characteristics of crops of winter wheat varieties infected with pathogens of leaf diseases.

In my opinion, the study requires major revisions before being accepted for publication in Plants.

The methodology and discussion of data must be reviewed and improved.

-Provide a flowchart of the main steps involved in the methodology developed in study.

-insert detailed information about the spectrometer used in the measurements (excitation source, collection, detector, ...)

-insert characteristic spectra (350 to 2500 nm) for the main foliar diseases detected in the study, and compare with that obtained for a healthy plant.

-evaluate the use of the ratio between spectrum channels to obtain correlations and verify the degree of disease development. For example, the reasons: 660\2035, 700\2035, 575\700, among others. The use of the channel ratio can better highlight statistical differences and correlations with diseases in cultivars.

-insert references in the discussion, for example, lines 353 to 366

-Why was it not evaluated to obtain a model for the varieties in the three years of studies (2021 to 2023)? (lines 449 to 456)

Comments on the Quality of English Language

Minor editing of English language required

Author Response

The study by Danilov et al (Plants-3072092), evaluated the use of ground-based spectrometry (range of electromagnetic radiation from 350 to 2500 nm) to verify the characteristics of crops of winter wheat varieties infected with pathogens of leaf diseases.

In my opinion, the study requires major revisions before being accepted for publication in Plants.

The methodology and discussion of data must be reviewed and improved.

-Provide a flowchart of the main steps involved in the methodology developed in study.

Response: According to this comment, a flowchart of the research methodology was added to the text of the manuscript. Lines 508-509.

-insert detailed information about the spectrometer used in the measurements (excitation source, collection, detector, ...)

Response: According to the comment, information about the device has been added. Lines 547-550.

-insert characteristic spectra (350 to 2500 nm) for the main foliar diseases detected in the study, and compare with that obtained for a healthy plant.

Response: Graphs of characteristic spectra have been added to the text of the manuscript. Lines 127-132.

-evaluate the use of the ratio between spectrum channels to obtain correlations and verify the degree of disease development. For example, the reasons: 660\2035, 700\2035, 575\700, among others. The use of the channel ratio can better highlight statistical differences and correlations with diseases in cultivars.

Response: We were interested in the possibility of carrying out such an analysis, but in a limited period of time we did not have time to select the appropriate ratios of spectral channels and find a methodological justification for them. We are willing to conduct such an analysis at a subsequent stage of reviewing this manuscript. If you have the opportunity to point us to relevant methodological sources, we would be grateful.

-insert references in the discussion, for example, lines 353 to 366

Response: As noted, links have been inserted into the discussion section. Lines 355, 358, 378, 381, 384, 457, 462, 475, 477.

-Why was it not evaluated to obtain a model for the varieties in the three years of studies (2021 to 2023)? (lines 449 to 456)

Response: The main goal of our research was a detailed assessment of the influence of variety factors and disease development on the spectral characteristics of crops of three varieties of winter wheat in the growing seasons of three years of research. We did not set a task to evaluate the variety model. The discussion section mentions the variety model in the context of prospects for further development of research. Without a doubt, the data obtained will become the basis for building a predictive model for diagnosing the development of pathogens.

Reviewer 2 Report

Comments and Suggestions for Authors

This manuscript explores the effects of varietal factors and disease development on the spectral characteristics of winter wheat. The background and process of the study are described in detail, and the work is somewhat novel. Below are my suggestions for improvement:

1. Consider rewriting excessively long sentences. Additionally, check grammar and paragraphs for verbosity.

2. The abstract suggests additional data to support the conclusions.

3. In "2. Results" need to add spectral graphs for comparison and analysis.

4. In "3. Discussion", it is proposed to add subheadings to discuss different aspects so that readers can read them clearly.

5. What are the limitations and future prospects of the study? Please reflect this in the discussion.

6. In "4. Materials and Methods", it is suggested that a map of the spatial location of the study area be added.

7. Reduce similarity (Percent match: 26%).

Comments on the Quality of English Language

Consider rewriting excessively long sentences. Additionally, check grammar and paragraphs for verbosity.

Author Response

This manuscript explores the effects of varietal factors and disease development on the spectral characteristics of winter wheat. The background and process of the study are described in detail, and the work is somewhat novel. Below are my suggestions for improvement:

  1. Consider rewriting excessively long sentences. Additionally, check grammar and paragraphs for verbosity.

Response: According to this note, the text of the manuscript has been revised.

  1. The abstract suggests additional data to support the conclusions.

Response: Changes have been made to the annotation in accordance with the comments.

  1. In "2. Results" need to add spectral graphs for comparison and analysis.

Response: Graphs of characteristic spectra have been added to the text of the manuscript. Lines 127-132.

  1. In "3. Discussion", it is proposed to add subheadings to discuss different aspects so that readers can read them clearly.

Response: In keeping with this note, the discussion section has been divided into subheadings. Lines 338, 386, 418, 455.

  1. What are the limitations and future prospects of the study? Please reflect this in the discussion.

Response: Limitations and prospects of the research are reflected in subsection 3.4 “discussion”. Lines 455-477.

  1. In "4. Materials and Methods", it is suggested that a map of the spatial location of the study area be added.

Response: A map of the spatial location of the study area has been added to the text of the manuscript. Lines 489-492.

  1. Reduce similarity (Percent match: 26%).

Response: According to this comment, work was carried out to change the text of the manuscript.

Reviewer 3 Report

Comments and Suggestions for Authors

Recommendations to authors

General comments:

- All tables need to be improved to be in accordance with the template of the journal.

- Add pictures, if possible, to show the impact of pathogen development.

- Add graphs using spectral reflectance curves of leaf surface reflectance at different stages of pathogen development or for different leaf diseases. Please consider replacing some tables with graphs and moving the tables to an appendix.

- Discussion is lengthy; please consider making it more concise.

Detailed comments:

- In lines 2 – 3: Remove the quotation marks from the title.

- In line 89: Is it possible to add some pictures that demonstrate the impact of pathogen development at several stages or percentages? I understand that the study focuses on the statistical analysis of the data 

- In line 99:  Make changes and ensure that the table caption format adheres to the journal's template, i.e. font should be smaller. Also: the p in p-values must be italic. Replace all commas (,) with periods (.) in numbers. I suggest removing p-values from Table 1 and leaving the asterisk to the values to show that they are statistically significant. This would enhance the readability of the table. Please also consider making a replacement: instead of "* - mathematically reliable influence of the factor on SBC indicators has been confirmed" just note that "* statistically significant". 

- In line 125: Check the format of the caption. Table 2 needs to be improved; see the comments above for the changes that need to be made. Superscript letters, rather than a separate column, should indicate groups of data based on statistical significance; please make the changes needed. The year ‘2021’ is bold, while ‘2022’ and ‘2023’ are not.

- In line 163: check my comments above for the changes that needed to be made to Table 3.

- In line 225: In Table 4: Superscript letters (just beside each number), rather than a separate column, should indicate groups of data based on statistical significance.

- In line 244: Table 5: same comments as above.

- In line 258: Table 6: same comments as above.

- In line 280: Table 7: same comments as above.

- In line 294: Table 8: same comments as above.

- In line 321: Table 9: same comments as above.

- In line 325: Table 10: Replace all commas (,) with periods (.) in numbers.

- In line 345: Please avoid the use of “trend”. It is better to use the word 'pattern'. The term 'trend' is typically used for long-term, extensive data.

- In lines 358 – 360: Please consider replacing the sentence with “Simultaneously, variability at other wavelengths within the spectral range depended on the composition and disease intensity of the pathogenic background”. The English in the discussion section needs improvement in several parts.

- In line 490: The eye-based calculation of the ratio of the proportion of the affected area of the plant leaf blade to its total area suggests that there is bias to the process of identifying the disease manifestation degree. Maybe you should use an algorithm or random forests to classify the leaves to clusters based on the percentage of the leaf damage. I also believe that you should mention in the text the limitations of the proposed methodology. 

Author Response

General comments:

- All tables need to be improved to be in accordance with the template of the journal.

Response: As noted, all tables have been revised.

- Add pictures, if possible, to show the impact of pathogen development.

Response: Relevant images have been added. Lines 531-533.

- Add graphs using spectral reflectance curves of leaf surface reflectance at different stages of pathogen development or for different leaf diseases. Please consider replacing some tables with graphs and moving the tables to an appendix.

Response: Response: Graphs of characteristic spectra have been added to the text of the manuscript. Lines 127-132.

- Discussion is lengthy; please consider making it more concise.

Response: The text of the discussion has been finalized and divided into subheadings.

Detailed comments:

- In lines 2 – 3: Remove the quotation marks from the title.

Response: Quotes from the title have been removed.

- In line 89: Is it possible to add some pictures that demonstrate the impact of pathogen development at several stages or percentages? I understand that the study focuses on the statistical analysis of the data 

Response: We have added images demonstrating the influence of pathogen development in the “materials and methods” section (lines 531-533.). The images show the symptoms of both individual pathogens and variants with the simultaneous manifestation of several diseases on one leaf blade. Unfortunately, at the moment, for objective reasons, we are not able to present the entire list of pathogens with different gradations of development. For example, yellow rust and powdery mildew are missing from the image. It is not possible to supplement the image data now since the wheat has already passed the growing season.

- In line 99:  Make changes and ensure that the table caption format adheres to the journal's template, i.e. font should be smaller. Also: the p in p-values must be italic. Replace all commas (,) with periods (.) in numbers. I suggest removing p-values from Table 1 and leaving the asterisk to the values to show that they are statistically significant. This would enhance the readability of the table. Please also consider making a replacement: instead of "* - mathematically reliable influence of the factor on SBC indicators has been confirmed" just note that "* statistically significant". 

Response: The table has been modified in accordance with the comment. Lines 103-107.

- In line 125: Check the format of the caption. Table 2 needs to be improved; see the comments above for the changes that need to be made. Superscript letters, rather than a separate column, should indicate groups of data based on statistical significance; please make the changes needed. The year ‘2021’ is bold, while ‘2022’ and ‘2023’ are not.

Response: The table has been modified in accordance with the comment. Lines 136-141.

- In line 163: check my comments above for the changes that needed to be made to Table 3.

Response: The table has been modified in accordance with the comment. Lines 173-176.

- In line 225: In Table 4: Superscript letters (just beside each number), rather than a separate column, should indicate groups of data based on statistical significance.

Response: The table has been modified in accordance with the comment. Lines 238-241.

- In line 244: Table 5: same comments as above.

Response: The table has been modified in accordance with the comment. Lines 256-259.

- In line 258: Table 6: same comments as above.

Response: The table has been modified in accordance with the comment. Lines 268-271.

- In line 280: Table 7: same comments as above.

Response: The table has been modified in accordance with the comment. Lines 287-290.

- In line 294: Table 8: same comments as above.

Response: The table has been modified in accordance with the comment. Lines 301-304.

- In line 321: Table 9: same comments as above.

Response: The table has been modified in accordance with the comment. Lines 322-325.

- In line 325: Table 10: Replace all commas (,) with periods (.) in numbers.

Response: The table has been modified in accordance with the comment. Lines 333-336.

- In line 345: Please avoid the use of “trend”. It is better to use the word 'pattern'. The term 'trend' is typically used for long-term, extensive data.

Response: This comment has been taken into account and corrections have been made to the text.

- In lines 358 – 360: Please consider replacing the sentence with “Simultaneously, variability at other wavelengths within the spectral range depended on the composition and disease intensity of the pathogenic background”. The English in the discussion section needs improvement in several parts.

Response: According to this comment, work was carried out to change the text of the manuscript.

- In line 490: The eye-based calculation of the ratio of the proportion of the affected area of the plant leaf blade to its total area suggests that there is bias to the process of identifying the disease manifestation degree. Maybe you should use an algorithm or random forests to classify the leaves to clusters based on the percentage of the leaf damage. I also believe that you should mention in the text the limitations of the proposed methodology. 

Response: The visual accounting method is a fundamental way to diagnose the degree of manifestation of external signs of the development of fungal diseases. Classical methods of phytosanitary monitoring of industrial crops are based on it. In addition, it is used in immunological studies of wheat varieties. For these purposes, there are generally accepted scales for assessing the degree of disease development [1-3]. Of course, this method is not without a certain amount of subjectivity, but initially it is possible to assess the degree of disease development in experimental crops only by eye. In our case, all three years of research were carried out by one specialist plant pathologist.

  1. Peterson. R.F.; Cambell. A.B.; Hannah. A.E. A diagrammatic scale for estimating rust intensity on leaves and stems of cere-718 als.Can. J. Res. 1948. 26, 496–500. 719
  2. Roelfs. A.P.; Singh. R.P.; Saari. E.E. Rust Diseases of Wheat: Concepts and Methods of Disease Management, CIMMYT: 720 Mexico City, Mexico, 1992. ISBN 968612747X. 721
  3. Eyal. Z.; Charen. A.L.; Prescott. J.M. van Ginkel. M. The Septoria Diseases of Wheat: Concepts and Methods of Disease 722 Management; CIMMYT: Mexico City, 1987. Mexico.

Round 2

Reviewer 1 Report

Comments and Suggestions for Authors

The questions were satisfactorily answered.

Author Response

Thank you for your attention to our work.

Reviewer 3 Report

Comments and Suggestions for Authors

Recommendations to authors

- In line 238: I think that Table 4 (as well as Figure 5 in line 256), regarding the results of a posteriori analysis of the spectral characteristics of winter wheat crops with different gradations of disease development, could be moved to the supplementary section, and instead, a graph depicting these values can be presented. If figure 1 represents these values, then you can safely move the tables to supplementary.

- In line 420: a chapter should never begin with "However, …".

- In line 490: Figure 2 (line 490) can be smaller in size. 

- In line 510: Figure 3 has low resolution, and it should be replaced. Also, boxes and lines in this figure are not aligned and therefore suggest that the figure was constructed somewhat hastily. Please improve Figure 3. 

- In line 533: Figure 4 is a nice addition to the text. I suggest that you place both photos horizontally. The text below the photos is distorted; better move the explanation to the caption. 

- The discussion section is still too lengthy. Several paragraphs can be summarized and shortened. Adding sub-sections was a good idea, but some modifications are still needed; try to reduce the text by highlighting only the most important points.

- In line 591: Please replace “A regular and clear division of the spectral characteristics of winter wheat by disease intensity was recorded only at a pathogen development level of over 5%.” with " A regular and clear distinction in the spectral characteristics of winter wheat by disease intensity was observed only when the pathogen development level exceeded 5%".

- In line 601: Check the repetition with line 592.

Author Response

Comments and Suggestions for Authors

Recommendations to authors

- In line 238: I think that Table 4 (as well as Figure 5 in line 256), regarding the results of a posteriori analysis of the spectral characteristics of winter wheat crops with different gradations of disease development, could be moved to the supplementary section, and instead, a graph depicting these values can be presented. If figure 1 represents these values, then you can safely move the tables to supplementary.

Response: We have transferred all tables with the results of post-hoc analysis to the appendices. In our opinion, Figure 1 fully reflects the spectral characteristics of the studied varieties over three years of research.

- In line 420: a chapter should never begin with "However, …".

Response: This comment has been taken into account and appropriate corrections have been made to the text.

- In line 490: Figure 2 (line 490) can be smaller in size. 

Response: The comment has been taken into account.

 - In line 510: Figure 3 has low resolution, and it should be replaced. Also, boxes and lines in this figure are not aligned and therefore suggest that the figure was constructed somewhat hastily. Please improve Figure 3. 

Response: Figure 3 has been modified.

- In line 533: Figure 4 is a nice addition to the text. I suggest that you place both photos horizontally. The text below the photos is distorted; better move the explanation to the caption. 

Response: The drawing has been modified according to the comment.

- The discussion section is still too lengthy. Several paragraphs can be summarized and shortened. Adding sub-sections was a good idea, but some modifications are still needed; try to reduce the text by highlighting only the most important points.

Response: In accordance with the recommendation, the text of the “Discussion” section was optimized in meaning and shortened.

- In line 591: Please replace “A regular and clear division of the spectral characteristics of winter wheat by disease intensity was recorded only at a pathogen development level of over 5%.” with " A regular and clear distinction in the spectral characteristics of winter wheat by disease intensity was observed only when the pathogen development level exceeded 5%".

Response: The appropriate replacement has been made.

- In line 601: Check the repetition with line 592.

Response: Repetition checked.
